# Synergy in Cystic Fibrosis Therapies: Targeting SLC26A9

**DOI:** 10.3390/ijms222313064

**Published:** 2021-12-02

**Authors:** Madalena C. Pinto, Margarida C. Quaresma, Iris A. L. Silva, Violeta Railean, Sofia S. Ramalho, Margarida D. Amaral

**Affiliations:** BioISI—Biosystems & Integrative Sciences Institute, Faculty of Sciences, University of Lisboa, Campo Grande, C8, 1749-016 Lisboa, Portugal; mdcpinto@fc.ul.pt (M.C.P.); mcquaresma@fc.ul.pt (M.C.Q.); iasilva@fc.ul.pt (I.A.L.S.); vrailean@fc.ul.pt (V.R.); ssramalho@fc.ul.pt (S.S.R.)

**Keywords:** SLC26A9, anion channels, CFTR, Cl^−^ secretion, Cystic Fibrosis

## Abstract

SLC26A9, a constitutively active Cl^−^ transporter, has gained interest over the past years as a relevant disease modifier in several respiratory disorders including Cystic Fibrosis (CF), asthma, and non-CF bronchiectasis. SLC26A9 contributes to epithelial Cl^−^ secretion, thus preventing mucus obstruction under inflammatory conditions. Additionally, SLC26A9 was identified as a CF gene modifier, and its polymorphisms were shown to correlate with the response to drugs modulating CFTR, the defective protein in CF. Here, we aimed to investigate the relationship between SLC26A9 and CFTR, and its role in CF pathogenesis. Our data show that SLC26A9 expression contributes to enhanced CFTR expression and function. While knocking-down SLC26A9 in human bronchial cells leads to lower wt- and F508del-CFTR expression, function, and response to CFTR correctors, the opposite occurs upon its overexpression, highlighting SLC26A9 relevance for CF. Accordingly, F508del-CFTR rescue by the most efficient correctors available is further enhanced by increasing SLC26A9 expression. Interestingly, SLC26A9 overexpression does not increase the PM expression of non-F508del CFTR traffic mutants, namely those unresponsive to corrector drugs. Altogether, our data indicate that SLC26A9 stabilizes CFTR at the ER level and that the efficacy of CFTR modulator drugs may be further enhanced by increasing its expression.

## 1. Introduction

Cystic Fibrosis (CF), a life-shortening genetic disorder affecting ~90,000 individuals worldwide [1], is caused by mutations in the CFTR (Cystic Fibrosis Transmembrane Conductance Regulator) gene [2], which encodes a cAMP-regulated chloride (Cl^−^) and bicarbonate (HCO_3_^−^) channel expressed at the apical plasma membrane (PM) of epithelial cells [3]. Defective CFTR leads to an imbalance in salt and fluid transport that results in dehydrated and viscous secretions. CF is a multi-organ disease, affecting the intestine, pancreas, and reproductive tract, but the lungs are the most critically affected organ. In fact, lung failure is the main cause of mortality and morbidity in individuals with CF [4,5,6].

More than 2100 CFTR genetic variants have been described so far, with only less than 20% (~360 variants) having been confirmed as disease-causing mutations [7,8]. These different mutations originate distinct cellular defects, including protein misfolding and aberrant PM trafficking (class II, which includes the most common mutation, F508del), reduced or absent channel function, lower PM stability, and diminished or even absent CFTR expression [9,10]. Recently, a new highly effective modulator therapy (HEMT) named Trikafta [11,12] has made a significant change in the lives of people with CF with at least one copy of F508del-CFTR, altogether representing ~80–85% of all individuals with CF [11,12]. This drug combines two CFTR correctors—VX-661 and VX-445—improving CFTR trafficking to the PM, and one potentiator—VX-770—increasing CFTR open channel probability. Nevertheless, the remaining 15–20% do not respond to this HEMT nor to other currently available modulators (e.g., VX-809) [10], making the search for alternative therapies a priority in CF research.

A popular alternative therapeutic strategy has been the activation of other (non-CFTR) channels/transporters, as it would apply to any individual with CF, irrespective of the CFTR mutations, i.e., a “mutation-agnostic” approach. An example of one such potential therapeutic target is Solute carrier family 26 member 9 (SLC26A9), a member of the SLC26 family of multifunctional anion transporters [13]. SLC26A9 functions as a constitutively active Cl^−^ transporter [14] and is predominantly expressed in epithelial lining the airways, stomach, duodenum, ileum, and pancreas [15,16], thus holding the potential to compensate for the absence of functional CFTR. Moreover, besides this potential role, SLC26A9 was reported to act as a CF gene modifier, with some of its polymorphisms and expression levels influencing disease severity and response to CFTR-modulator drugs [17,18,19]. SLC26A9 polymorphisms were linked to an increased risk of developing meconium ileus and pancreatic disease in people with CF [20,21,22], and higher expression of SLC26A9 correlates to a delayed manifestation of CF-related diabetes [23]. Furthermore, reduced expression of SLC26A9 is linked to asthma in children [24]. Moreover, previous studies showed that siRNAs targeting SLC26A9 cause a decrease in wt-CFTR function [25], and proposed that SLC26A9 expression favors the biogenesis and/or stabilization of CFTR, resulting in turn, in enhanced cAMP-activated currents [26].

The exact direct role that SLC26A9 plays in CF and in other respiratory disorders is, however, yet to be elucidated. Although it is described as a Cl^−^ transporter, its localization was found to be mainly cytoplasmic or at the tight junctions (TJ) level [27]. It was also proposed to interact with both wild-type (wt) and F508del-CFTR, being its PM expression strongly dependent on CFTR [25,28]. Namely, SLC26A9 appears to be retained in the endoplasmic reticulum (ER) when co-expressed with F508del-CFTR, which leads to the degradation of both proteins by the proteasome [27].

Altogether, these data not only highlight the impact of SLC26A9 expression in lung disease and response to CFTR modulator drugs but also motivated our research on the interaction between SLC26A9 and CFTR. Thus, our main goal here was to better understand the interplay between these two proteins, and how SLC26A9 might be used to develop novel CF therapies to be applied on their own or in combination with those currently available to further enhance their effects. Our data first confirmed that SLC26A9 and CFTR co-localize in human native lung tissue, including in CF epithelia where SLC26A9 expression is reduced. We then demonstrated that in native non-CF lung epithelia SLC26A9 localizes at the TJ level as well as in the sub-apical compartment. In human bronchial (CFBE) cells, we showed that SLC26A9 expression levels strongly affect CFTR expression and function. Particularly, F508del-CFTR rescue by HEMT is influenced by SLC26A9 expression. Interestingly, SLC26A9 overexpression did not significantly alter the PM expression of other CFTR traffic mutants, namely those that are not rescued by HEMT. This suggests a role for SLC26A9 in stabilizing CFTR at the ER, thus increasing the pool available to traffic along the secretory pathway and reach the PM. Taken together, these data support a role for SLC26A9 as a clinically relevant disease modifier and promising therapeutic target not only to circumvent deficient Cl^−^ secretion in CF but also to increase the effect of the already available CFTR modulator therapies.

## 2. Results

### 2.1. SLC26A9 and CFTR Show Distinct Expression Patterns in Control vs. CF Airway Tissues and Primary Cells

To address the interaction of SLC26A9 with CFTR in human airways, we first investigated their subcellular localization in native human airway tissue. To this end, we stained healthy control and F508del/F508del bronchi sections for SLC26A9, CFTR and tight junctions (TJ) marker zonula occludens-1 (ZO-1). In control tissue, SLC26A9 appears to co-localize with CFTR (Figure 1, arrows), although CFTR shows more discrete staining, whereas some SLC26A9 localizes diffusely in the sub-apical compartment of the epithelium. Similarly, SLC26A9 also seems to co-localize with ZO-1 (Figure 1, arrows), concentrating at the TJ, but also being present sub-apically. Interestingly, in CF tissue (F508del/F508del), SLC26A9 also appears to co-localize with F508del-CFTR and ZO-1 (Figure 1, arrowheads). However, as expected, F508del-CFTR shows faint and diffuse cytoplasmic staining (albeit sub-apical), as a consequence of dysfunctional CFTR and so does ZO-1 as a consequence of aberrant TJ [29,30]. Importantly, in CF tissue, SLC26A9 localization is also intracellular, and its expression levels appear to be decreased.

Thus, these data support the co-localization of SLC26A9 with CFTR and ZO-1 in human native airway tissue: either apically in non-CF tissue, or as a diffuse sub-apical localization in CF (F508del/F508del) tissue.

Based on this co-localization of CFTR and SLC26A9 with TJ protein ZO-1, we assessed how CFTR and SLC26A9 expression levels vary over the course of differentiation by Western Blot (WB) in primary human nasal epithelial (pHNE) cells derived from an individual with CF homozygous for the F508del mutation and from a non-CF control. In non-CF pHNE cells we observed that CFTR expression increased during differentiation, while SLC26A9 expression is constant over the course of differentiation, it still showed a trend towards higher expression levels after 7 and 14 days in ALI conditions (Appendix A). However, in F508del/F508del pHNE cells, although F508del-CFTR expression is strongly augmented on day 7 (same as wt-CFTR), SLC26A9 shows a trend towards reduced expression when compared to control pHNE cells (*p* = 0.01 for day 7 wt vs. F508del, Appendix A). Interestingly, there is a delay in SLC26A9 expression in F508del/F508del pHNE cells, being only increased on day 14 (Appendix A).

### 2.2. SLC26A9 Knockdown Decreases CFTR Expression and Function

Next, we aimed to confirm how reducing SLC26A9 expression influences CFTR expression, trafficking, function, and the response to HEMT, i.e., the combination of correctors tezacaftor (VX-661) with elexacaftor (VX-445). We stably transduced CFBE cells expressing wt- or F508del-CFTR (see Section 4) with a specific shRNA targeting SLC26A9 (or shLuciferase [shLuc] as a negative control). After confirming SLC26A9 knockdown (KD) by both WB and semi-quantitative PCR (Figure 2A,C and Appendix A, respectively), we assessed CFTR protein levels by WB and observed a significant decrease in wt-CFTR expression upon SLC26A9 KD (Figure 2A,B). Regarding F508del-CFTR, SLC26A9 KD led to a significant decrease in its rescue by VX-661 + VX-445 in comparison to shLuc-treated cells (Figure 2A,B), as observed by the lower amount of fully glycosylated CFTR (Band C), indicating a reduced expression at the PM. Moreover, when comparing the fold increase in band C between conditions exposed or not to VX-661 + VX-445 for shLuc and shSLC26A9, the former (under shLuc) is substantially higher (~9 and ~6, respectively).

We then evaluated the effect of SLC26A9 KD on CFTR function by Ussing chamber. Despite that, SLC26A9 KD was modest in CFBE wt-CFTR cells (~31%, Figure 2C), cAMP-activated currents (i.e., CFTR-mediated currents) were significantly inhibited (~65%) when compared to shLuc-treated cells (Figure 3A,C). Likewise, in CFBE F508del-CFTR cells, although the negligible currents of DMSO-treated cells were not significantly changed by SLC26A9 KD, its rescue by VX-661 + VX-445 was reduced by ~35% (Figure 3B,C).

To confirm the effects of SLC26A9 KD on CFTR expression and function in a system expressing endogenous CFTR, we then stably transduced the 16HBE cell lines expressing either wt- or F508del-CFTR with the same shRNAs targeting either SLC26A9 or Luciferase (control) and assessed the effect of all the clinically available CFTR correctors VX-809, VX-661 and VX-445. As expected, total wt- and F508del-CFTR protein expression were significantly reduced, and the rescue of F508del-CFTR PM expression (band C) by VX-445 alone, or in combination with VX-661, was also diminished (Appendix A). SLC26A9 KD in 16HBE cells expressing wt-CFTR also originated lower cAMP-activated currents (Appendix A), and both the untreated and VX-809-treated F508del-CFTR expressing cells showed lower CFTR activity (Appendix A).

These data suggest that SLC26A9 KD (confirmed in Appendix A) decreases CFTR PM expression and function in both endogenous and overexpression settings.

### 2.3. SLC26A9 Overexpression Potentiates CFTR Expression and Function

Given the negative impact of SLC26A9 KD on wt-CFTR expression and function, and importantly, on F508del-CFTR rescue by the clinically available modulators, our next step was to examine the effect of increasing SLC26A9 levels on F508del-CFTR mediated Cl^−^ currents and protein expression. To this end, we stably transfected CFBE F508del-CFTR cells with a GFP-tagged SLC26A9 construct, under the control of an inducible (Tet-On) promoter system.

After adding doxycycline (Dox) to induce the SLC26A9 construct expression in F508del-CFTR CFBE cells, we could detect SLC26A9 (Figure 4A,C) by WB. Notably, since in this WB exposure times were very short (due to high expression levels of the inducible labeled construct) endogenous SLC26A9 was not detected here (in contrast to Figure 2A) due to its significantly lower expression levels. SLC26A9 overexpression, and still without VX-661 + VX-445, still led to the appearance of the fully glycosylated form of F508del-CFTR (Band C), albeit in very small amounts (Figure 4A,B). Remarkably, in SLC26A9 overexpressing cells, the rescue of F508del-CFTR by VX-661 + VX-445 was much stronger when compared to control cells (−Dox) (Figure 4A,B). This greater rescue of F508del-CFTR by VX-661 + VX-445 in the presence of higher levels of SLC26A9 was further corroborated by the significantly increased cAMP-activated currents obtained in Ussing chamber experiments (Figure 5A,B).

Similar data were obtained in 16HBE cells overexpressing SLC26A9 thus further validating these results and demonstrating that SLC26A9 enhances total and PM expression of endogenous wt-CFTR (Appendix A) and rescue of F508del-CFTR by all the correctors tested (VX-809, VX-661, VX-445 and the combination of VX-661 with VX-445) (Appendix A). These data are of particular relevance, as we show here that the clinically approved therapies for people carrying the F508del mutation can be improved by increasing SLC26A9 expression.

Interestingly, lowering SLC26A9 expression (by KD through shRNA transfection) resulted in a delay in cell polarization of CFBE wt-CFTR cells, as seen by the lower values of transepithelial resistance (TEER) on day 3, eventually reaching the levels of the control cells on day 5 (Appendix A). No differences were observed in CFBE F508del-CFTR cells (Appendix A). However, CFBE F508del-CFTR cells overexpressing SLC26A9 showed a significant increase in TEER after Dox induction (Appendix A). Remarkably, CFBE cells transduced with the inducible SLC26A9 construct have—even without adding Dox—a higher TEER when compared to non-transfected cells or transfected with control shRNAs. This result might be explained by the leaky expression of SLC26A9 in the absence of Dox (Appendix A and Appendix A).

### 2.4. Overexpression of SLC26A9 Does Not Alter the PM Expression of other CFTR Traffic Mutants

Since F508del is classified as a class II mutation, i.e., has impaired trafficking [6], we decided to investigate the impact of SLC26A9 overexpression in cells expressing other class II CFTR mutations, namely G85E, R560S, and N1303K (Figure 6). None of these CFTR mutants was rescued by VX-661 + VX-445, as observed by the absence of band C, i.e., mature form of CFTR (Figure 6) after treatment with these drugs. Similarly, SLC26A9 overexpression did not alter the expression levels of any of the mutants, although there is a trend to slightly higher CFTR expression levels for these mutants upon SLC26A9 overexpression (Figure 6). Thus, these results indicate that SLC26A9 contributes to increasing CFTR levels at the PM only when it is able to exit the ER, namely wt-CFTR and F508del-CFTR when rescued by HEMT. Notwithstanding SLC26A9 overexpression seems to contribute to some ER-retained stabilization, at least for F508del-CFTR and possibly also some other class II mutants. Notably, it was demonstrated that SLC26A9 interacts with CFTR bearing other rare mutations, being its expression also enhanced after the rescue of these mutants by CFTR modulators [31].

## 3. Discussion

In the present work we demonstrated the synergistic effect of SLC26A9 expression levels on CFTR expression and function, and most importantly, on the rescue of F508del-CFTR by the HEMT currently available in the clinic (VX-661 + VX-445). Consistent with previous data [27], our data show that SLC26A9 and CFTR seem to co-localize in native human bronchial epithelia, either in samples from a non-CF individual (control) at the PM or intracellularly in tissues from an individual with CF (F508del/F508del).

Although thus far, the co-localization of SLC26A9 and CFTR has been assessed mostly through their heterologous expression either in HEK293T or CFBE41o-cells transfected with tagged SLC26A9 constructs [25,26,28], recently the same was also shown in well-differentiated primary human bronchial epithelial (pHBE) cell cultures [27]. However, here we show for the first time that SLC26A9 and CFTR also appear to co-localize in native human bronchial epithelial CF (F508del/F508del) and non-CF tissue. Native bronchial tissues provide a unique insight into the pathological and molecular events of the disease and are good snapshots of the physiological situation. These data provide evidence of SLC26A9 and CFTR co-localization in human native lung epithelial (CF and healthy control) are in line with previous studies reporting that SLC26A9 and CFTR physically interact, both its wt and F508del forms [14,26]. Our results are also in agreement with studies showing that SLC26A9 expression levels are reduced in CF [27].

Importantly, it was established that SLC26A9 and CFTR behave differently in polarized vs. non-polarized cells [25], namely at the level of TJs, and that the negative effect of F508del-CFTR on SLC26A9 is more pronounced in well-differentiated epithelial cells endogenously expressing these proteins [27], evidencing the importance of studying physiologically relevant model systems. Here, we also observed that in control lung epithelia SLC26A9 localizes to the TJs as well as in the sub-apical compartment, as previously reported in pHBE cells [27]. However, we also show that in CF epithelia, the TJ protein ZO-1 shows intracellular and diffuse staining and, although SLC26A9 and ZO-1 still appear to co-localize, the integrity of TJ structures is compromised. This is in line with our previous findings (both in lung tissue and immortalized CFBE cells) demonstrating that the presence of mutant CFTR occurs with a shift in the localization of ZO-1 from a TJ/apical to a more intracellular one [32]. These data are also consistent with multiple reports of abnormal TJ formation caused by mutant CFTR [29,30,33], as well as with the proposal that functional CFTR interacts directly with ZO-1 to regulate TJ formation [29].

Also consistent with these data on diffuse SLC26A9 expression at the compromised TJ structures in CF [32], our results in CF (F508del/F508del) pHNE cells also suggest a delayed expression of SLC26A9 over the course of differentiation vs. control cells. Moreover, our data indicate that lower SLC26A9 expression levels may affect cell polarization. Indeed, these data show, similarly to those by other authors [27], that SLC26A9 KD resulted in a delay in cell polarization in CFBE wt-CFTR cells, as seen by a decrease in TEER values. In parallel, SLC26A9 KD had no effect on the TEER of CFBE F508del-CFTR cells, likely due to the fact that CF cells already have compromised TJs and low TEER values [32]. However, CFBE F508del-CFTR cells overexpressing SLC26A9 showed a significant increase in TEER, suggesting that a higher expression of SLC26A9 may potentiate cell polarization.

To further investigate the relationship between SLC26A9 and CFTR we used two different cell models: CFBE, which stably overexpress CFTR; and 16HBE, with endogenous CFTR expression. We took advantage of molecular biology tools to decrease (shRNA) or increase (cDNA) SLC26A9 expression levels in these cells. Consistently in both systems, SLC26A9 KD led to a reduction in wt- and F508del-CFTR protein expression and in CFTR-mediated Cl^−^ transport. F508del-CFTR rescue by the clinically available HEMT was also inhibited in both cell models by SLC26A9 KD. On the other hand, SLC26A9 overexpression increased F508del-CFTR rescue by HEMT (expression and function), confirming that it is possible to further improve the current best treatment for people with CF carrying F508del. Enhancing SLC26A9 expression could therefore increase Cl^−^ secretion in two ways: (i) by its own function as a Cl^−^ transporter (and therefore bringing benefits for everyone with CF, regardless of their genotype), and (ii) by its synergistic rescuing effect with the HEMT VX-661 + VX-445.

Given ours and previous data showing the effect of SLC26A9 expression in PM-localized CFTR (either wt, corrected F508del, or G551D [17]) and the failure to rescue other trafficking mutants (G85E, R560S and N1303K), we hypothesize that SLC26A9 plays a role in stabilizing some misfolded CFTR mutants at the ER which are then available to be rescued to the PM by modulators that act on these mutants. This hypothesis is consistent with previous reports which already suggested stabilization of CFTR by SLC26A9 in the biogenesis/maturation process [26]. Furthermore, the effect of SLC26A9 on CFTR stability appears to be rather specific, as SLC26A9 overexpression does not alter the expression levels of other membrane proteins such as Na^+^/K^+^ ATPase (data not shown).

Importantly, despite that some class II mutations tested here (namely G85E and N1303K) showed a significant—albeit modest—correction by Trikafta in studies in primary nasal epithelial cultures from CF patients [34], we acknowledge the differences and limitations of immortalized cells and overexpressing systems. Nonetheless, we show here that the results in CFBE cells (overexpressing CFTR) were reproducible, and even more significant, in 16HBE cells which have an endogenous expression of CFTR. In our view, the results obtained here and by other authors [34] indicate that SLC26A9 might actually improve the response to the correctors in people with CF carrying F508del-CFTR and possibly other class II mutations that respond to these drugs. Nonetheless, more studies are needed to confirm this hypothesis. An interesting approach would be to assess the effect of SLC26A9 overexpression on other trafficking mutations known to respond better to the treatment with Trikafta, such as M1101K [34], or overexpressing SLC26A9 in primary cells.

In summary, our data revealed that SLC26A9 expression is crucial for enhanced CFTR PM expression and function, while also having a potential effect on TJ structures. Moreover, increasing SLC26A9 levels is a potential therapeutic option for CF, by providing an additional source of Cl^−^ secretion, and/or by enhancing the effect of the already clinically available HEMT.

## 4. Materials and Methods

### 4.1. Cell Culture

Immortalized Human Bronchial Epithelial cells (16HBE41o-) endogenously expressing wt-CFTR (16HBE wt-CFTR) or genetically modified to express F508del-CFTR (16HBE F508del-CFTR), and Human Embryonic Kidney (HEK) 293T were grown in Minimum Essential Medium (MEM) supplemented with 10% fetal bovine serum (FBS). Cystic Fibrosis Bronchial Epithelial cells (CFBE) expressing wt, F508del, G85E, R560S or N1303K-CFTR were grown in MEM supplemented with 10% FBS and 2.5 µg/mL puromycin. These cells were generated by lentiviral transduction of CFBE 41o- (parental) cells. Primary human nasal epithelial (pHNE) cells were obtained from nasal brushing of healthy individuals or people with CF with an F508del/F508del genotype. Nasal brushings were obtained from Hospital Santa Maria and Hospital Dona Estefânia (Lisbon, Portugal) after receiving patients’ written consent and approval by the hospital Ethics Committee. After isolation, cells were expanded and differentiated according to protocols established by Jeffrey Beekman’s lab (Utrecht, The Netherlands) [35]. Briefly, after expansion, cells were seeded in a porous membrane and cultured under Air-Liquid Interface (ALI) conditions for 14 days. Protein was collected after 0, 7, 14 days of differentiation in ALI conditions. All cells were cultured at 37 °C in a humidified atmosphere of 5% (*v*/*v*) CO_2_.

### 4.2. Native Human Lung Tissue

Explanted CF lungs (F508del/F508del) and healthy control tissues were collected in the Paediatrics Department of Motol University Hospital (Prague, Czech Republic) under approval of applied regulations and the hospital’s Ethics Committee and shipped over 24 h to Lisboa. Informed consent was obtained from all subjects. The explanted lungs were dissected in order to identify the secondary and tertiary bronchi by removing the connective tissue using a scalpel and surgical scissors. The bronchi were then washed with cold PBS several times to remove the mucus and cut into small fragments followed by overnight fixation with electron microscopy grade PFA (0.2% *v*/*v*, Electron Microscopy Sciences, 15710) and then dehydrated and slowly frozen. After lung cleaning and fixation, the pieces of secondary/tertiary bronchi were kept for 12 h at a time in phosphate buffers with increasing sucrose (Fluka, 84100) content (4% to 15%, *w*/*v*) and then incubated in a final solution with 15% (*w*/*v*) sucrose and 7.5% (*w*/*v*) gelatine (Sigma-Aldrich, G9391) for 1 h at 37 °C. Dry ice-chilled isopentane (VWR, 24872) was then used to slowly freeze the tissues, which were kept at −80 °C until sectioning. Tissue sections were cryocut using a Leica CM1850 UV cryostat. Cryosections 15–30 μm thick were generated on Superfrost^®^ Plus slides (Thermo Scientific, 10149870), and used right away for immunohistochemistry, or stored at −20 °C until further use.

### 4.3. Immunofluorescence (IF) Lung Tissue Staining

Lung tissue sections on Superfrost^®^ Plus slides were permeabilized with triton X-100 (Amersham Biosciences, 17-1315-01) 0.2% (*v*/*v*), after which tissue autofluorescence was quenched with NaBH_4_ (1 mg/mL, Sigma-Aldrich, 213462). A blocking step with BSA 1% (*w*/*v*) was performed before incubating overnight at 4 °C with primary antibodies. The following primary antibodies were used: rabbit polyclonal anti-SLC26A9 (BioTechne, NBP2-30425), mouse monoclonal anti-CFTR (CFF, 570), and mouse monoclonal anti-ZO-1 (Invitrogen, 33-9100). The following day a mix of the secondary antibodies (anti-mouse Alexa 488 and anti-rabbit Alexa 568, Life Technologies, A21202 and A10042) and nuclear dye (4 μg/mL, Methyl Green, Sigma-Aldrich, 67060) was applied for 2 h at room temperature (RT). Filter sections were mounted in a mix of N-propylgallate (Sigma-Aldrich, P3130) and Glycerol for microscopy (Merck, 104095). The tissues stained were secondary/tertiary bronchi and were as similar as possible for comparison. Areas of extensive shedding/remodeling in CF tissue were avoided in the analysis, and areas of intact epithelia preferred. Imaging was performed with a Leica TCS SP8 confocal microscope coupled to a Hamamatsu Flash4 sCMOS camera, using an HC Plan Apo 20×/0.75 objective. Software used for acquisition was Leica’s LAS x, and image processing was performed on ImageJ FIJI [36]. FIJI was used to generate maximum image projections (MIPs).

### 4.4. Lentiviral Transduction

HEK 293T cells were used to produce lentiviral particles containing shRNAs targeting either SLC26A9 or Luciferase (shLuc) as a control (MISSION^®^ shRNA from Sigma, Germany) or the cDNA of GFP-tagged SLC26A9 in a Tet-On 3G inducible expression system. HEK cells were seeded at a density of 5 × 10^5^ cells per well of a poly-lysine coated 6-well plate and incubated for 24 h at 37 °C, 5% CO_2_ (*v*/*v*). Then, cells were transfected with 5 µg of DNA per well—2.38 µg of packaging plasmid pCMV-dR8.74psPAX2, 0.24 µg of enveloping plasmid VSV-G/pMD2.G, and 2.38 µg of the plasmid containing the shRNA/cDNA. The cells were then incubated for 18 h at 37 °C, 5% CO_2_. The medium was replaced to remove the transfection reagent and the cells were incubated for 30 h at 37 °C, 5% CO_2_. The media containing the lentiviral particles were harvested and the packaging cells were discarded. The harvested viral particles were immediately used to transduce CFBE or 16HBE cells seeded the day before on 6-well plates. Cells were infected with 2 mL of lentivirus-containing medium, plus 4 µg/mL of Polybrene (Hexadimethrine bromide, (Sigma-Aldrich, H9268)) infection enhancer. The plates were centrifuged at 200× *g* for 1 h at 25 °C and then incubated for 24 h at 37 °C, 5% CO_2_. The medium was then changed to the original cell medium supplemented with selection antibiotics (2.5 μg/mL puromycin for shRNA transfected cells, and 2.5 μg/mL puromycin + 500 μg/mL G418 for SLC26A9 overexpressing cells) to eliminate the non-transduced cells. In the case of SLC26A9 overexpressing cells, 1 μg/mL doxycycline (Dox) was added 48 h before experiments to induce its expression.

### 4.5. RT-PCR

Semi-quantitative reverse-transcriptase (RT)-PCR was performed to quantify SLC26A9 gene knockdown. Total RNA was isolated using the NZY Total RNA Isolation kit (NZYtech, Portugal). Total RNA (1 μg/20 μL reaction) was reverse-transcribed using random primers and NZY M-MuLV Reverse Transcriptase (NZYtech, Portugal). Each RT-PCR reaction contained sense and antisense primers for SLC26A9 (0.5 µM) or for GAPDH (0.5 µM), 0.5 μL cDNA and NZYTaq II DNA polymerase (NZYtech, Portugal). After 2 min at 95 °C cDNA was amplified during 30 cycles for 30 s at 95 °C, 30 s at 56 °C and 1 min at 72 °C. PCR products were visualized by loading on RedSafe Nucleic Acid Staining Solution (Intron Biotechnology) containing agarose gels and analyzed using Image Lab (BioRad).

### 4.6. Western Blotting

Protein extracts from CFBE or 16HBE cells were separated on 8.5% (*w*/*v*) polyacrylamide gels and transferred into polyvinylidene difluoride (PVDF) membranes. Membranes were blocked with 1 or 5% (*w*/*v*) non-fat milk powder (NFM) in Tris buffer saline with Tween 20 (TBS-T) or in PBS-T for 1 h at RT and incubated overnight at 4 °C with primary antibodies (rabbit polyclonal anti-SLC26A9 (BioTechne, NBP2-30425) diluted 1:250 in 1% NFM/PBS-T or mouse CFTR 596 antibody (Cystic Fibrosis Foundation (CFF), 1:3000 in 5% NFM/TBS-T). Membranes were also probed with 1:3000 mouse anti-α-tubulin in 5% NFM/PBS-T (Sigma, T5168) as loading control. The membranes were incubated with HRP-conjugated goat anti-rabbit/anti-mouse IgG (diluted 1:3000 in 1% or 5% NFM/TBS-T or NFM/PBS-T) for 2 h at RT. Chemiluminescent detection was performed using the Clarity™ Western ECL substrate (BioRad, 170-5061) and the Chemidoc™ XRS system (BioRad). The quantification of band intensity was performed using the Image Lab software (BioRad) and normalized to the loading control as appropriate.

### 4.7. Ussing Chamber

CFBE or 16HBE cells were seeded at approximately 2.5 × 10^5^ cells/mL or 3.5 × 10^5^ cells/mL, respectively, onto 12 mm Snapwell™ inserts with 0.4 µm pore polyester membrane (Corning, 3801) and 1.12 cm^2^ area. Transepithelial electrical resistance (TEER) of the monolayers was measured with a chopstick electrode (STX2 from WPI^®^, Berlin, Germany) and electrophysiological analyses were carried out in monolayers with resistance values around or above 600 Ω × cm^2^. Transepithelial resistance (R_te_) was determined by applying 1 s current pulses of 0.5 µA (5 s-period). For Ussing chamber measurements, Snapwell inserts were mounted in the chamber device and continuously perfused with Ringer containing (mM): NaCl 145, K_2_HPO_4_ 1.6, MgCl_2_ 1, KH_2_PO_4_ 0.4, Ca-Gluconate 1.3, Glucose 5, pH 7.4. Low Cl^−^ ringer (NaCl 30 mM) was added to the luminal side to create an electrochemical driving force. Experiments were performed in the presence of Amiloride (20 μM), an inhibitor of the Epithelial Sodium (Na^+^) Channel (ENaC). CFTR was activated by the adenylyl cyclase activator Forskolin (2 µM) and Genistein (25 µM) added to the luminal side and inhibited by the specific inhibitor CFTR inhibitor 172 (30 µM). Values for the transepithelial voltage (V_te_) were referenced to the luminal epithelial surface. Equivalent short-circuit currents (I_eq-sc_) were calculated according to Ohm’s law from V_te_ and R_te_ (_Ieq-sc_ = V_te_/R_te_), with appropriate correction for fluid resistance.

### 4.8. Statistical Analyses

Data are reported as means ± SEM. Student’s *t*-test (for paired or unpaired samples as appropriate) or ANOVA were used for statistical analysis. A value of *p* ≤ 0.05 was accepted as statistically significant. The number of experiments performed (*n*) is indicated in figure legends.

## Figures and Tables

**Figure 1 ijms-22-13064-f001:**
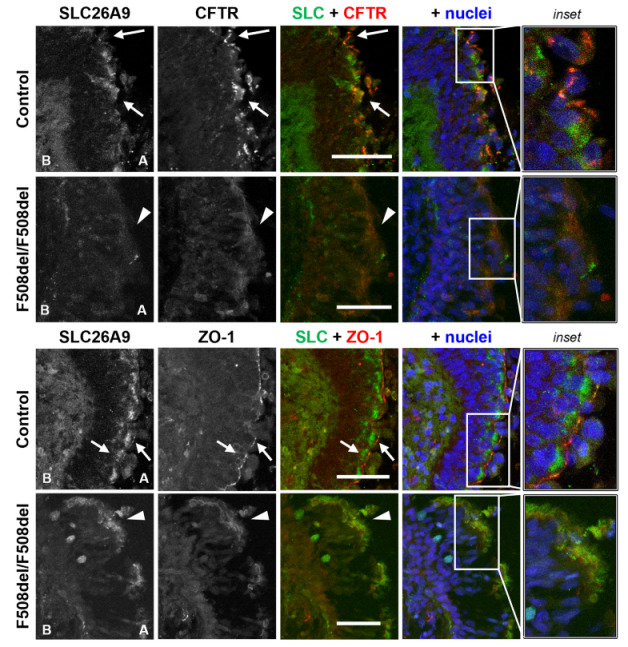
SLC26A9 has a sharp apical TJ-like localization in control lung but a faint (apical) cytoplasmic localization in F508del/F508del lung. Representative images of native human bronchial tissue (Control and F508del/F508del) immunostained for SLC26A9, CFTR and ZO-1. Nuclei are depicted in blue, SLC26A9 in green and CFTR/ZO1 in red. Scale bars represent 50 μm. Images are displayed as maximum image projections (MIPs). Apical and basal (A and B, respectively) sides of the epithelia are identified. SLC26A9 co-localizes with CFTR and ZO-1, whether they show apical localization (arrows, control) or diffuse cytoplasmic expression (arrowheads). Close-ups of the areas identified with arrows/arrowheads (insets) are shown on the right-hand side. The stained tissues display secondary/tertiary bronchi and were as similar as possible for comparison. Areas of extensive shedding/remodeling in CF tissue were avoided, and areas of intact epithelia preferred (*n* = 3 sections).

**Figure 2 ijms-22-13064-f002:**
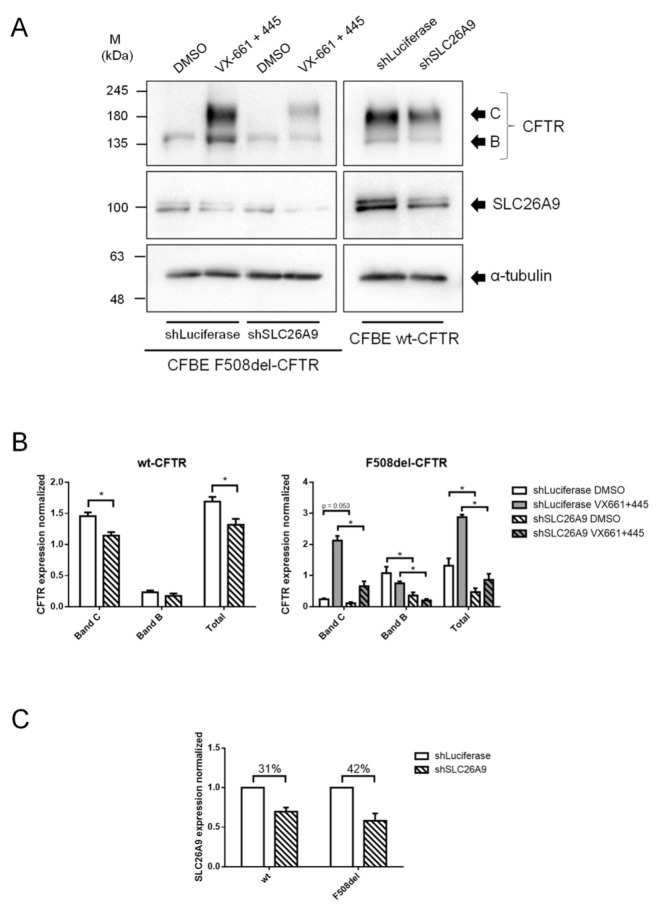
SLC26A9 knockdown decreases both wt- and F508del-CFTR expression and F508del-CFTR correction by VX-661 + VX-445. (**A**) Western Blot showing CFTR and SLC26A9 expression in shLuciferase vs. shSLC26A9 transduced cells. α-tubulin was used as a loading control. (**B**) Quantification by densitometry of CFTR protein expression detected by WB in (**A**). From left to right: Band C, which corresponds to fully glycosylated (PM-localized) CFTR; Band B, the core-glycosylated form of CFTR; and total protein expression. (**C**) Quantification of SLC26A9 knockdown, showing a decrease of ~31% and ~42% of SLC26A9 expression in wt-CFTR and F508del-CFTR CFBE cells, respectively, when compared to shLuciferase-transduced cells. Data are normalized to loading control and showed as arbitrary units, mean ± SEM (number of experiments (*n*) = 3). Statistical analyses were performed by GraphPad Prism 6.0 using unpaired t-test where “*” indicates statistically significant differences (*p* ≤ 0.05, unpaired *t*-test) between cells transduced with shLuciferase and shSLC26A9.

**Figure 3 ijms-22-13064-f003:**
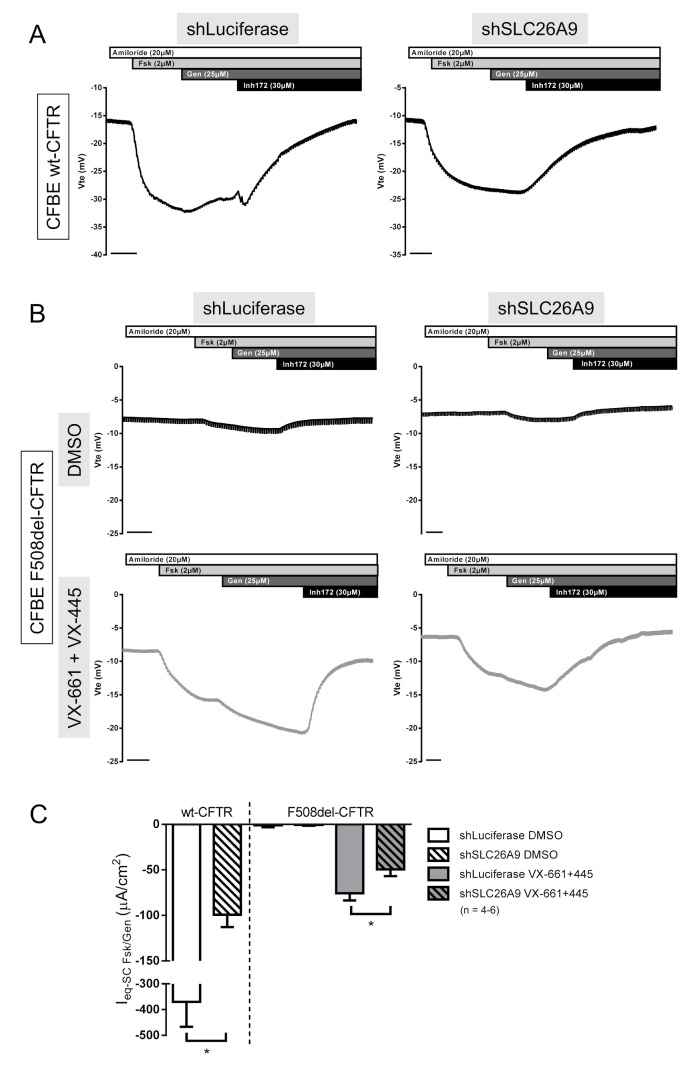
SLC26A9 knockdown inhibits wt-CFTR function and the correction of F508del-CFTR by VX-661 + VX-445. (**A**) Original Ussing chamber tracings obtained for CFBE wt-CFTR cells transduced with the control shRNA shLuciferase (left) or shSLC26A9 (right). CFTR was activated by 2 μM Forskolin (Fsk) and 25 μM Genistein (Gen) in the presence of the epithelial sodium channel (ENaC) inhibitor (Amiloride, 20 μM) and was inhibited by CFTR-inhibitor 172 (Inh172, 30 μM). All compounds were added to the luminal side. (**B**) Original Ussing chamber tracings obtained for cAMP-induced Cl^−^ currents activated and inhibited as in (**A**) in the absence (top tracings, black) or in the presence (lower tracings, grey) of 5 μM VX-661 + 2 μM VX-445 for CFBE F508del-CFTR cells transfected with shLuciferase (left) or shSLC26A9 (right); (**C**) Summary of I_sc-eq_ currents of CFBE wt-/F508del-CFTR stably transduced with shLuciferase vs. shSLC26A9 and treated with DMSO vs. VX661 + VX-445. Data are represented by mean ± SEM and “*” indicates statistically significant differences (unpaired *t*-test, *p* ≤ 0.05). The number of filters (*n*) used in the statistical analyses is indicated in the graph. Scale bars in (**A**,**B**) represent 1 min.

**Figure 4 ijms-22-13064-f004:**
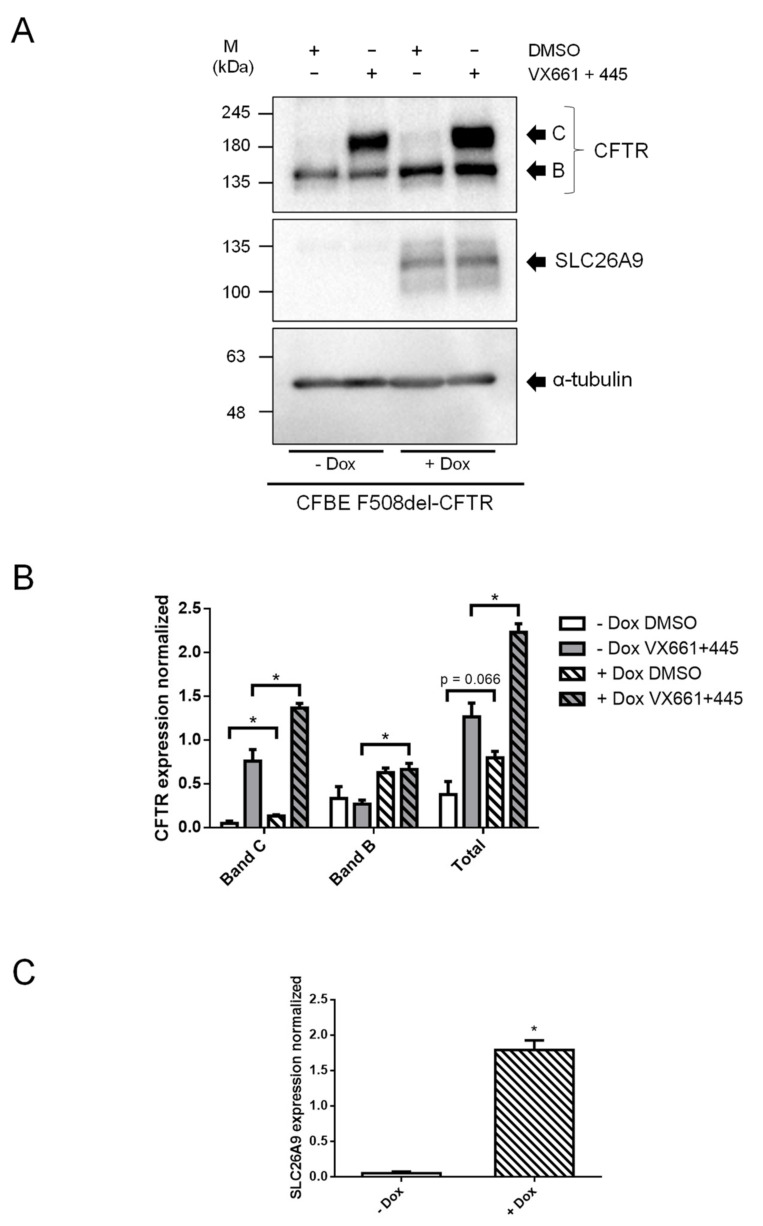
SLC26A9 overexpression increases F508del-CFTR expression and correction by VX-661 + VX-445. (**A**) Western Blot showing CFTR and SLC26A9 expression in CFBE F508del-CFTR cells stably transduced with SLC26A9 in a Tet-On system, where SLC26A9 is only expressed after Doxycycline induction (+Dox). α-tubulin was used as a loading control. (**B**) Quantification by densitometry of CFTR protein expression detected by WB in (**A**). (**C**) Quantification of SLC26A9 expression after Dox induction, as shown in (**A**). Data are normalized to loading control and showed as mean ± SEM (*n* = 3). (*) indicates significant differences between cells in the absence (−Dox) or presence (+Dox) of doxycycline, the latter meaning overexpression of SLC26A9 (unpaired *t*-test, *p* ≤ 0.05).

**Figure 5 ijms-22-13064-f005:**
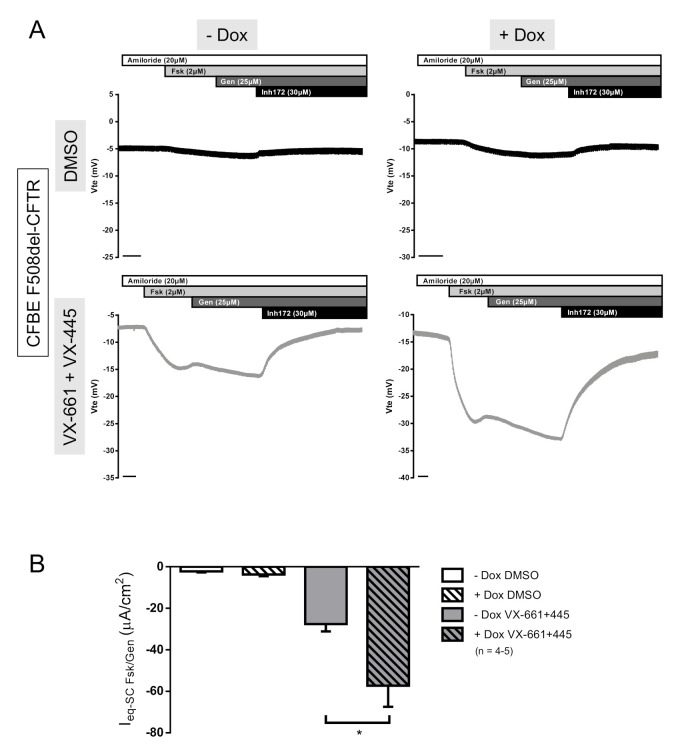
SLC26A9 overexpression heightens F508del-CFTR correction by VX-661 + VX-445. (**A**) Original Ussing chamber tracings obtained for cAMP-induced Cl^−^ currents (Fsk + Gen) in the absence (top tracings, black) or in the presence (lower tracings, grey) of 5 μM VX-661 + 2 μM VX-445 for CFBE F508del-CFTR cells with (right) or without (left) SLC26A9 overexpression (+Dox and −Dox, respectively); (**B**) Summary of I_sc-eq_ currents of CFBE F508del-CFTR +/− Dox and treated with DMSO vs. VX661 + VX-445. Data are represented by mean ± SEM. (*) indicates significant statistical difference between cells treated or untreated with Dox (unpaired *t*-test, *p* ≤ 0.05). The number of filters (*n*) used in the statistical analyses is indicated in the graph. Scale bars in (**A**) represent 1 min.

**Figure 6 ijms-22-13064-f006:**
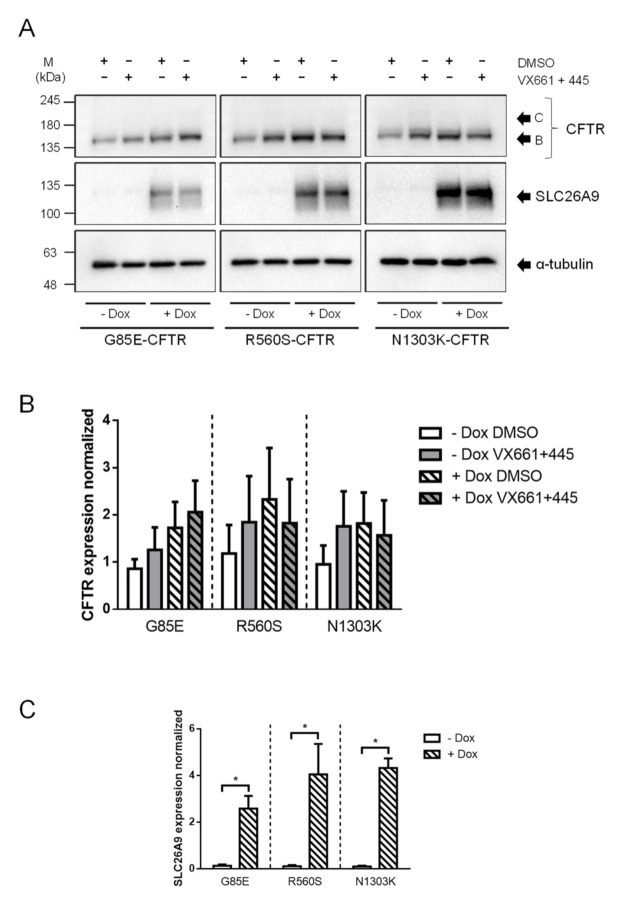
Overexpression of SLC26A9 does not alter the PM expression of other class II CFTR mutants. (**A**) Western Blot showing CFTR and SLC26A9 expression in CFBE cells transduced with SLC26A9 and stably expressing one of the following class II CFTR mutants: G85E, R560S, or N1303K. SLC26A9 expression is detectable after Doxycycline induction (+Dox). α-tubulin was used as a loading control. (**B**) Quantification of CFTR protein expression detected by WB in (**A**). (**C**) Quantification of SLC26A9 overexpression after Dox induction, as shown in (**A**). Data are normalized to loading control and showed as mean ± SEM (*n* = 3). (*) indicates significant difference between cells in the absence or presence of Dox (unpaired *t*-test, *p* ≤ 0.05).

## Data Availability

Not applicable.

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
