# Peer review of "Synergy in Cystic Fibrosis Therapies: Targeting SLC26A9"

_ijms, 2021, doi:10.3390/ijms222313064_

Round 1

Reviewer 1 Report

In their article entitled “Synergy in Cystic Fibrosis therapies: targeting SLC26A9”, Pinto et al present promising new results supporting the potential therapeutical interest of targeting SLC26A9 when treating patients with CF. While of great interest, some points have to be addressed before this manuscript becomes acceptable for publication.

Major comments:

In figure 1, co-localization of SLC26A9 and CFTR/ZO-1 should be quantified, if possible, or further evidenced (e.g. using PLA) in order to strengthen the conclusions made by the authors. Similarly, the decrease in SLC26A9 expression is not obvious when comparing for example the first control (upper panel, left) and the second F508-del/F508-del condition (bottom panel, left). Some quantification would be helpful, as well as Western Blots and PCR results on lung cells supporting this difference between patients.

Page 4, the authors say that “However, in F508del/F508del pHNE cells, although F508del-CFTR expression is strongly augmented on day 7 (same as wt-CFTR), SLC26A9 clearly shows a trend towards reduced expression when compared to control pHNE cells (Figure 1C,D)”. This sentence is a bit confusing: according to figure 1S, in both wt-CFTR and F508del-CFTR pHNE cells, there is no significant increase in SLC26A9 between day 0 and day 7, or even between day 0 and day 14 for wt-CFTR. This last result is also in apparent contradiction with the authors statement that “In non-CF pHNE cells we observed that CFTR expression increased during differentiation, an increase that was also accompanied by SLC26A9 expression (Figure S1A,B)”. Some clarification is therefore required.

According to the authors, “Regarding F508del-CFTR, SLC26A9 KD led to a significant decrease in its rescue by VX-661 + VX-445 in comparison to shLuc-treated cells (Figure 2A,B)”. However, Figure 2B shows that while indeed the total amount of CFTR is reduced, the fold increase between conditions exposed or not to VX-661 + VX-445 for shLuc and shSLC26A9 could be similar. Can this result be solely explained by the initial decrease in CFTR expression induced by shSLC26A9?

Throughout the article, the authors try to prove that the expression of SLC26A9 controls CFTR expression in the plasma membrane of several cell models. These results could be strengthened thanks to extra experiments such as biotinylation, immunofluorescence (combined with membrane staining), …etc.

If the SLC26A9 overexpression experiments are carried out in the same cells used in Figure 2, why no endogenous expression of SLC26A9 is visible in figure 4A without doxycycline? Did the authors use an anti-GFP antibody?

The authors final conclusion that SLC26A9 is “crucial not only for proper CFTR PM expression and function but also for TJ structure” (page 12) seems premature. Indeed, they do not present evidence in the manuscript that changes in SLC26A9 expression and/or function are directly responsible for TJ structure. This conclusion should therefore be either amended or supported by specific experiments.

Reviewer 2 Report

In this paper, the authors studied the relationship between two chloride channels, CFTR and SLC26A9. In bronchial tissues of non-CF and CF patients, they analyzed the localization/colocalization of the two channels. Using cell lines, they showed a regulation of the WT-CFTR and F508del-CFTR expression, function and response to CFTR correctors by SLC26A9. The authors concluded that SLC26A9 expression is crucial for tight junction structure and cell polarization, and enhanced the efficacy of CFTR correctors.

While some of the results are unquestionable, the reviewer has concerns with other results that are found unconvincing or insufficient to support the conclusions.

Major concerns

1) The quality of the immunostaining of FIG. 1 is too low to conclude on the colocalization of CFTR and SLC26A9, either at the TJ level or in the cell cytoplasm. This is probably due to the poor preservation of non-CF and CF bronchial tissues.

The immunostainings appear like background noise. Is it possible to see the negative controls of the experiments?

Importantly, in CF tissue, SLC26A9 localization is also intracellular, and its expression levels are decreased”: analysis of protein expression by immunohistochemistry is not quantitative.

2) In figure S1, how is it possible to show graphs with error bars and statistics if the number of different tissues used is of 1 for CF as well as non-CF experiments?

What does “n = 3 experiments” mean if the culture of only one patient was analyzed?

It is difficult to conclude on a mechanism with the analysis of the culture of a single patient since the great inter-patient variability in ALI cultures of human respiratory cells is well described.

3) In the western blots of SLC26A9, we sometimes see one band, sometimes two. What is the specific band? Which band is quantified?In addition, one can wonder about the fact that there is no SLC26A9 in the cultures of CFBE-F508del-CFTR not treated with the dox (no band in Figure 4) whereas SLC26A9 is clearly seen in CFBE-F508del-CFTR transfected with ShLuciferase (Figure 2).Figure 4 and Figure S5 : Did the authors examine the expression of SLC26A9 and CFTR in CFBE-F508del-CFTR cells not transfected but treated with Dox vs not treated with Dox (as a control of their experiment)?

4) It is not possible to conclude to a crucial role of SLC26A9 in cell polarization and for TJ structure with a simple measure of TEER.  

Minor concerns

5) Figure 2 : It would have been interesting to compare the expression of SLC26A9 in the non-CF and CF cells, that is to say in the CFBE-F508del-CFTR cells without DMSO or alternatively in the CFBE-WT-CFTR cells in the presence of DMSO.

6) Tracings of Figure 3A are not representative of results in figure 3C because there is only a modest decrease in ShSLC26A9 cells

7) Figure S4 panel C : is there a significant difference between F508del-CFTR shLuciferase DMSO and F508del-CFTR shLuciferase VX-809? In other words, is there a CFTR rescue?

8) There is no description of the G85E, R560S and N1303K cells, anywhere in the manuscript.

9) Material and methods section : How native human tissues were processed is unclear.

Reviewer 3 Report

In the manuscript by Pinto et al., the authors elegantly demonstrated the role of SLC26A9 on CFTR functional expression at the plasma membrane. Moreover, they suggest the SLC26A9 modulation as a potential therapeutic option for CF. The manuscript is well written and is interesting, however I do have some suggestions:

Major concerns:

  • The authors investigated the expression of SLC26A9 after Trikafta treatment on rare CFTR mutations that are not responding to Trikafta. This reviewer strongly suggests to perform the same experiments with a rare mutations that are responding to Trikafta.

Minor concerns:

  • Page 2: “It was also proposed to interact with both wild-type (wt) and F508del-CFTR, being its PM expression strongly dependent on CFTR”. Please include the following citation: 10.1002/jcp.22967
  • Page 10: “Thus, these results indicate that SLC26A9 contributes to increasing CFTR levels at the PM only when it is able to exit the ER, namely wt-CFTR and F508del-CFTR when rescued by HEMT”. Recently it has been demonstrated that SLC26A9 interacts with other rare mutations and its expression is enhanced after the rescue of mutated CFTR with CFTR modulators (10.1016/j.jcf.2020.07.015).
  • Figure 6A: How the authors interpret their data such that the SLC26A9 expression is CFTR mutation dependent? In figure 6A they show a higher expression of SLC26A9 in N1303K or R560S-CFTR CFBE cells compare to F508del and G85E. Please clarify

Round 2

Reviewer 1 Report

No further comment

Reviewer 2 Report

I thank the authors for theirs answers and their clarifications. 

Reviewer 3 Report

No more concerns